

# A hybrid deep learning framework for skin disease localization and classification using wearable sensors

Xiaoling Zhao, Huixin Zhang, Qian Zheng and Caihong Jing

School of Nursing and Inner Mongolia Medical University, Hohhot, China

## ABSTRACT

Accurate detection of skin diseases is essential for timely intervention and treatment. This article proposes a patch-based, interpretable deep learning framework for skin disease detection using wearable sensors and clinical data. Specifically, a fully convolutional residual neural network (FCRN) is employed to extract local features from high-resolution skin images captured *via* wearable sensors, using a patch-level training approach. Pre-processing techniques—including image resampling, intensity normalization, and noise reduction—standardize the input data to ensure consistency across sensor variations. To enhance local feature learning, the FCRN incorporates residual modules, which mitigate gradient vanishing and improve model performance. The framework generates disease probability maps that visualize regions of high diagnostic risk, providing interpretable insights into skin anomalies. In the proposed methodology, a convolutional neural network (CNN) integrates image-derived features with clinical data such as patient demographics, symptoms, and medical history. This CNN-based multimodal fusion approach improves the model's ability to capture spatial relationships and enhances classification performance. Experimental evaluations demonstrate that the proposed framework achieves state-of-the-art results across multiple evaluation metrics, including accuracy, sensitivity, and specificity. The interpretable disease probability maps highlight affected skin regions, enhancing model transparency and clinical usability. This approach demonstrates the potential of combining wearable sensor technology with deep learning for efficient, scalable, and explainable skin disease detection, laying the foundation for real-time clinical applications.

# INTRODUCTION

The skin, the human body's largest organ, is a crucial barrier. The skin's primary role is to safeguard the human body against harmful external substances while preventing the loss of essential nutrients within (*Hameed et al., 2016*). The skin's structure includes the outer layer known as the epidermis, the underlying dermis, and the deeper subcutaneous tissues. It senses external factors and protects internal organs and tissues from detrimental bacteria, pollutants, and sunlight. The 20 to 23 square feet of skin surface area is critical in

Corresponding author
Caihong Jing,
caihingjingrenas12@outlook.com

thermoregulation, sensation, and overall protection (*ALEnezi, 2019*; *Mohammed & Al-Tuwaijari, 2021*). A variety of factors, such as exposure to solar radiation, smoking habits, alcohol consumption, physical activities, viral infections, and the surrounding work environment, influence the health of the skin. The interplay of these factors undermines skin function, results in negative health consequences, and, in extreme instances, poses a risk to human life (*Hameed, Shabut & Hossain, 2018*). Skin diseases have emerged as a prevalent issue affecting human health, carrying considerable implications for society and individuals. External influences, such as pollution, allergens, microorganisms, and internal elements, like genetic disorders, hormonal imbalances, or immune system problems, can lead to these outcomes (*Yang et al., 2018*).

Skin diseases are typically classified into three main types: viral, fungal, and allergic. While fungal and allergic conditions can often be treated effectively when diagnosed early, viral skin diseases require timely identification for proper intervention (*ALEnezi, 2019*). Infectious disorders, bacterial infections, and contact dermatitis are also prevalent, altering the skin's colour, consistency, and integrity (*Mohammed & Al-Tuwaijari, 2021*). Globally, skin diseases account for approximately 1.79% of physical disabilities, affecting 30% to 70% of individuals across different populations (*Yang et al., 2018*). Traditional methods for monitoring skin health and diagnosing diseases often involve invasive techniques, such as intravenous catheterization, large-scale ECG monitors, or plasma osmolality measurements. These approaches can be costly, time-consuming, and inconvenient, making them unsuitable for long-term, real-time monitoring (*Alaejos et al., 2019*). Medical devices such as pacemakers and implantable sensors have been used primarily in hospital settings for many years. However, wearable biosensors have revolutionised the healthcare landscape, offering a more accessible, non-invasive solution for health monitoring (*Ianculescu et al., 2018*; *Cheng et al., 2021*).

Wearable technology, first introduced in the mid-20th century, has evolved significantly. Modern wearable devices incorporate sensor components to monitor and record physical or biochemical parameters. These devices are versatile, integrated into clothing, adhered to the skin, or inserted into body orifices, and they facilitate continuous, real-time health monitoring outside clinical settings (*Ianculescu et al., 2018*; *Cheng et al., 2021*). In skin health management, wearable sensors can track vital parameters such as hydration levels using advanced technologies like electrodermal activity and wireless moisture sensors. This marks a shift from conventional invasive methods to user-friendly, real-time monitoring solutions (*Chun, Kim & Pang, 2019*). Such advancements address the limitations of traditional medical systems, which often face issues of limited resources and high costs, especially in underdeveloped regions. By enabling continuous health monitoring, wearable sensors can identify skin diseases at early stages, minimise risks, and improve patient outcomes (*Anbar, Gratt & Hong, 1998*). As skin diseases can be contagious and significantly impact the quality of life, timely diagnosis and treatment are critical to prevent further complications (*ALEnezi, 2019*).

This article proposes a multimodal deep learning framework for skin disease detection, combining patch-based feature extraction from wearable sensor images with clinical data fusion. The proposed methodology leverages a fully convolutional residual network

(FCRN) to extract localised spatial features from skin image patches and generate disease probability maps. These maps highlight high-risk regions, offering interpretability and aiding in disease localization. In addition, patient-specific clinical features, such as age, symptom duration, gender, and skin type, are fused with image-derived features using a convolutional neural network (CNN) for final classification. The main contributions of this study are as follows:

1. To improve detection accuracy and robustness, We propose a hybrid model integrating wearable sensor images (optical, thermal, and multispectral) with self-reported clinical data.
2. A FCRN is designed to extract localized features from image patches, enabling the generation of interpretable disease probability maps highlighting high-risk regions.
3. A CNN fuses spatial features from disease probability maps with patient-specific clinical data, improving the model's overall performance.

The remainder of the article is organized as follows: "Related Works" presents related works. "Methods and Materials" describes the proposed methodology, including patch-based feature extraction, disease probability map generation, and clinical data fusion. "Experimental Setup" reports the experimental setup, and "Result and Discussion" includes the results and discussion, including evaluation metrics, visualisations, and comparisons with baseline models. "Conclusion and Future Work" concludes the article and highlights future research directions.

## RELATED WORKS

The advancement of machine learning has addressed the limitations of conventional methods for diagnosing skin diseases, leading to the establishment of image recognition technology specifically tailored for this purpose. Identifying images through machine learning represents a convergence of various fields, including medical imaging of skin conditions, mathematical modelling, and computer technology. This integration utilises feature engineering and classification algorithms to recognise and diagnose skin diseases effectively. In 2012, *Garnavi, Aldeen & Bailey (2012)* employed wavelet decomposition to extract texture features, modelled and analysed lesion boundary sequences to obtain boundary features, and utilised shape indicators to derive geometric features. In conclusion, four distinct classifiers were employed for the classification task (*Garnavi, Aldeen & Bailey, 2012*). During the initial phase of diagnostic advancements, technology evolved as individuals developed tools capable of analysing nearly any desired analyte by collecting and transmitting samples to various laboratories. The second technological revolution of point of care testing (POCT) has now granted patients, nurses, and other medical professionals access to the lab. A recent technological advancement has surfaced, enabling patients to conveniently transport their test results through smart bio-monitoring wearable devices (*Song et al., 2021*). *Wong, Scharcanski & Fieguth (2011)* introduced an innovative iterative stochastic region-merging technique to segment skin lesion areas from macroscopic images. This method begins with stochastic region merging at the pixel level,

progressing to the region level until convergence is achieved (*Wong, Scharcanski & Fieguth, 2011*).

A distinct investigation indicates that over 99% of the resistance occurs at the surface level. Drawing from bioelectrical impedance analysis (BIA), we have collected galvanic skin response (GSR) data and created a method for assessing hydration levels using the GSR. GSR quantifies skin conductance, reflecting the inverse of skin resistance, and is evaluated through electrodermal activity (EDA). EDA sensors are employed to investigate sympathetic responses in humans. Nonetheless, it operates on the fundamental concept of transmitting light current from the human body at the skin's surface. Consequently, the GSR data from the EDA sensor is utilised through machine learning techniques to identify the hydration level. Nonetheless, the BIA presents a multifaceted approach unsuitable for monitoring purposes and is restricted to an indirect assessment of hydration level (HL) (*Fish & Geddes, 2009*). Classification involves a computational modelling endeavour in machine learning to predict a target category for a specific input data sample. The methodology initiates the forecast of the category labels of the given observations. Many times, the categories are marked as targets. If you have categorical input data and you want to forecast categorical output variables, you should try to estimate the modification matrix. Sorting incoming data into the correct category is the primary goal of categorization (*ExpertAI, 2022*).

*Reeder et al. (2019)* emphasize that these devices can precisely measure local sweat loss and chloride concentration during activities such as swimming and biking, showing a strong correlation with conventional methods. In one trial, athletes wearing the devices during intense physical activity received real-time feedback on their sweat composition and hydration status, allowing immediate adjustments to their hydration strategies (*Flament et al., 2021*). Techniques for extracting data from healthcare systems play a crucial role in the development of automated tools for disease diagnosis, leveraging both machine learning and deep learning algorithms. Researchers have employed various artificial intelligence algorithms to train classifiers essential for conducting machine diagnostics by applying machine learning and deep learning principles. The interconnectedness of artificial intelligence, machine learning, and deep learning represents an evolving journey. Utilizing advanced techniques to address data challenges and generate new accounts by leveraging extensive information facilitated by intelligent algorithms and conveyed through interconnected systems (*Liao, Li & Luo, 2016*).

In a separate study, the authors employed five machine-learning algorithms to identify skin diseases. Ultimately, the confusion matrix analysis revealed that the convolutional neural network model yielded the most favourable outcome for the disease detection process (*Bhadula et al., 2019*). Researchers have proposed an artificial intelligence system founded on a neural network. This system comprises two components. Initially, the process of image acquisition was utilized for feature extraction. The subsequent section involved categorization, executed through the feed-forward neural network (*Reddy & Nagalakshmi, 2019*). The research introduced a framework that uses images to diagnose skin conditions. The research categorised four distinct skin conditions: acne, cherry angioma, melanoma, and psoriasis, employing support vector machine (SVM), random

forest (RF), and k-nearest neighbors (KNN) methodologies. The collection comprised 377 images, with 80% designated for training and 20% allocated for testing. The approach involves using a medium filter for resizing images and eliminating noise. The images are subsequently transformed into greyscale. The technique developed by Otsu effectively differentiates among various diseases, while features are obtained through applying Gabor, entropy, and Sobel methods. The application of the RF algorithm yields an accuracy of 84.2%. The KNN algorithm achieved an accuracy of 67.1%. Nevertheless, the SVM classifier performs better than the others, attaining an accuracy of 90.7% (*Lefèvre-Utile et al., 2021*).

The field of artificial intelligence, known as machine learning, uses statistical and mathematical techniques to give computers human-like capacities. So, ML provides automation and improves machines' ability to learn from what they have seen without programming (*i.e.*, human intervention) (*Faggella, 2018*). Recently, advancements in machine learning and deep learning have shown great potential as valuable resources in dermatology. These methods provide automated and scalable approaches for diagnosing skin conditions, enhancing dermatologists' skills. Deep learning models, especially CNNs, have demonstrated remarkable potential in automatically diagnosing skin conditions. Models utilizing machine learning, when developed with a wide range of comprehensive datasets featuring images of skin diseases, have shown a remarkable ability to attain high accuracy and reliability in diagnostic processes—their potential lies in tackling the challenge of dermatologist shortages by providing accessible diagnostic tools. Nonetheless, the effectiveness of these models relies heavily on the quality and variety of the dataset employed for training and the clarity of the outcomes (*Sreekala et al., 2022*).

*Hajgude et al. (2019)* present a method for identifying 408 images related to eczema, impetigo, and melanoma skin conditions, along with a category designated for other images. The model is constructed utilizing various techniques: a median filter for noise reduction, the Otsu method for lesion segmentation, a 2D Wavelet transform for feature extraction such as entropy and standard deviation, and GLCM for extracting texture features including contrast and correlation. Applying SVM and CNN classifiers for disease classification yielded accuracy rates of 90.7% and 99.1%, respectively (*Hajgude et al., 2019*; *Mulge, 2024*; *Ahalya et al., 2024*).

The area of automation in skin disease classification has received attention with the deep learning boom (*Oztel, Yolcu Oztel & Sahin, 2023*) and constructed a portable diagnosis framework using CNNS for images captured from smartphones, though input RGB parameters constrained the depth of spatial features and diagnosis interpretation to mere 2D images. *Muhaba et al. (2022)* enhanced the hybrid system's diagnostic precision by integrating clinical data alongside image inputs, but their system underutilised local spatial features. *Sadik et al. (2023)* conducted an extensive review on the performance of different CNNs employing transfer learning, VGG16 and ResNet, integrating them with some primary school clinical data and wearable datasets, which were still absent in the literature. Armed with these data sets (*Chen et al., 2025*), spun attention mechanisms to features on pigmented lesion classification, which was still useful but added more value for claiming better analysis than whole-image followers. *Nassar et al. (2025)* shifted the

discussion to containing the explainability of the multi-class support deep classification framework towards sustaining healthcare, focused more on classifying diseases than features. Likewise, *Badr et al. (2024)* implemented a multi-model deep learning hierarchy for skin disease classification, but not for optimised deep edge wearable sensor-driven deployment.

The structure of the aerogel increases the sensor's responsiveness and compressibility to external pressure due to its lightweight and porous nature. The ability of this material to undergo piezoresistive changes makes it useful for measuring physiological signals such as heartbeat, breathing, muscle activity, and body position (*Zhang et al., 2024*; *Zhao et al., 2025b*). Treatment included topical and hydroxychloroquine, an antimalarial medication frequently used in autoimmune dermatological conditions, and systemic corticosteroids (*Wu et al., 2024*, *2025*). To reduce symptoms and the disruption of cancer treatment, early detection and prompt intervention are essential (*Zhao et al., 2025a*; *Wang et al., 2025*).

These studies (*Ahmed et al., 2021*) indicate significant advances in image-based diagnostics. However, most do not incorporate clinical reasoning or real-world implementation into their approaches. On the other hand, our proposed hybrid framework combines patch-based feature extraction from high-resolution images captured by wearable sensors with clinical data from the patient using multimodal fusion. This approach integrates clinical relevance with diagnostic precision and interpretability, addressing usability in practical settings. Although previous attempts at integrating AI in detecting skin diseases have shown some promise, they remain fundamentally flawed in ways that impact clinical accuracy and reliability. Most models still operate at the level of whole-image scrutiny and do not extract features at a localized level. This makes critical lesion-level details diffused and less meaningful, thus decreasing their interpretability. For instance, models employing standard CNN architectures such as VGG16 or ResNet fail to capture any delicate regional anomalies because of their global feature pooling strategies. In addition, most frameworks assume that image data is the only relevant data, completely ignoring the clinical context, which includes the patient's age, gender, medical history, and the severity of the symptoms, all of which play a pivotal role in forming a rational diagnosis. Systems that include clinical metadata tend to merge them too late and without sufficient meaningful feature interaction, leading to poor performance. Many models exhibit a lack of explainability, which further restricts the adoption of such systems into modern clinical practice since professionals need clear and logical reasons, both visually and contextually, for decisions made by an AI. Another limitation is the ability to use real-time applications since most other methods are intensive in the resources required to compute them, making them unsuitable for edge deployment, integration into wearables, or sensor use. All these gaps indicate the demand for a more context-aware, lightweight, interpretable framework.

## METHODS AND MATERIALS

This section describes the strategy and resources employed to detect skin diseases. The combined approach merges image-derived spatial features and clinical data into a multi-modal deep learning structure. The two data types used for this study were sourced

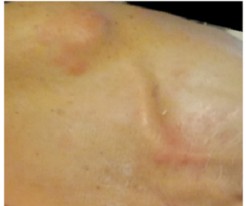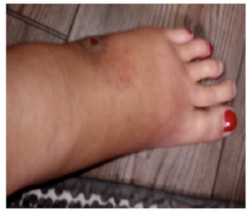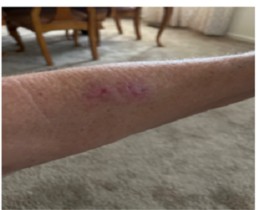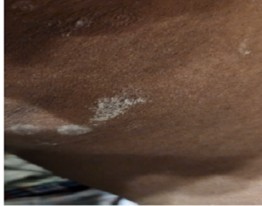

**Figure 1  Sample images from dataset.**

from the SCIN (Skin Condition Image Network) dataset. Advanced wearable sensors were used as the wearable device. Resampling, normalizing, and patching images were conducted in the preprocessing stages to ensure uniformity in the data.

## Third party dataset DOI/URL

This research integrates the SCIN (Skin Condition Image Network) Open Access Dataset (*Google Research Datasets and Stanford Medicine, 2024*), and data gathered using wearable sensors.

ULR of third-party dataset: DOI: 10.5281/zenodo.10819503.

The SCIN dataset hosted on the Google Cloud contains over 10,000 images of demographics such as eczema, psoriasis, acne, and melanoma uploaded by Google search users in America. Each contributor provided relevant self-reported demographic information, including age, gender, ethnicity, symptom history, and Fitzpatrick Skin Type (sFST) to ensure diversity and fairness. The sample images from the dataset can be seen in Fig. 1.

## Selection method

The proposed use of multimodal deep learning techniques for skin disease diagnosis devices enables the balance of the two conflicting features of interpretability and practical applicability. Furthermore, the diagnostic accuracy attained is outstanding. Therefore, WEN would significantly spend resources on improving the interpretability and practical applicability. It would be worth noting that the inclusion of clinical data was embedded using CNN, which is how patient data was fused with image features for enhanced accessibility to analysis. The generalised approach was utilised alongside disease-specific analysis which aided in overcoming baseline models. Such robustness ensured accuracy and a better AUC score, reflecting superiority over VGG16 and ResNet18. Taking this further into consideration, veb3 technology can also aid in machine tools that have better interpretability and clinical explanation—this would enable doctors to plan better strategies to combat disease. Real and real-time fluid interactions empower the doctors and give them the needed control. Future adaptation might include temporal and spatial tracking with the help of wearable sensors. Alongside CNN, additional measures such as fusion algorithms with real-time portable devices for improved AI explainability would further the transition and enhance the battle against skin diseases.

SCIN was bolstered by employing wearable sensor devices that enabled capturing social images and clinical data in great detail. The sensors used optical imaging for fine detail

images, thermal sensors for temperature variations, and multispectral sensors for structural and pigmentation. Additional metadata, such as demographics, symptom duration, and environmental data, enabled collecting over 10,000 additional images. This integrated dataset catered to high-quality skin images, clinical metadata, and real-time sensor measurements. The intended model will attain high accuracy and practicality for real-time skin disease detection due to this multi-modal dataset, and spatial features from skin images and clinical insights from metadata will be learned, enhancing accuracy and robustness.

### Data preprocessing

The preprocessing pipeline for skin disease images captured using wearable sensors ensures consistency, quality, and suitability for deep-learning models. To standardise the input image size, all images were resampled to a consistent spatial resolution of $256 \times 256$ pixels. Given an input image $I(x, y)$ with dimensions $H \times W$, the resampled image $I'(x, y)$ is computed as:

$$I'(x, y) = I\left(\frac{x \cdot H}{H'}, \frac{y \cdot W}{W'}\right), \quad \forall x \in [0, H'], \ y \in [0, W'] \tag{1}$$

where $H'$ and $W'$ are the target height and width, respectively. Spatial adaptive non-local means (SANLM) filtering was applied to preserve fine details while reducing sensor noise. The denoised image $I_d$ at pixel $p$ is computed as:

$$I_d(p) = \frac{\sum_{q \in N_p} w(p, q) I(q)}{\sum_{q \in N_p} w(p, q)}, \quad w(p, q) = \exp\left(-\frac{\|I(p) - I(q)\|^2}{h^2}\right) \tag{2}$$

where $N_p$ represents the neighborhood of the pixel $p$, $w(p, q)$ is the similarity weight, $I(q)$ is the intensity at the pixel $q$, and $h$ is a smoothing parameter. This technique adaptively smooths regions while preserving edges. Non-skin areas such as clothing or backgrounds were removed using a segmentation technique. A threshold-based mask $M(x, y)$ was generated:

$$M(x, y) = \begin{cases} 1 & I(x, y) > T(skinpixel) \\ 0, & otherwise \end{cases} \tag{3}$$

where $T$ is a threshold value determined through Otsu's method. The final segmented image $I_s$ was obtained by element-wise multiplication:

$$I_s(x, y) = I(x, y) \cdot M(x, y). \tag{4}$$

To ensure uniform brightness across all images, pixel intensities were normalised to a range of $[0, 1]$ using min-max scaling. For an input image $I(x, y)$, the normalised image $I_n(x, y)$ is computed as:

$$I_n(x, y) = \frac{I(x, y) - I_{min}}{I_{max} - I_{min}} \tag{5}$$
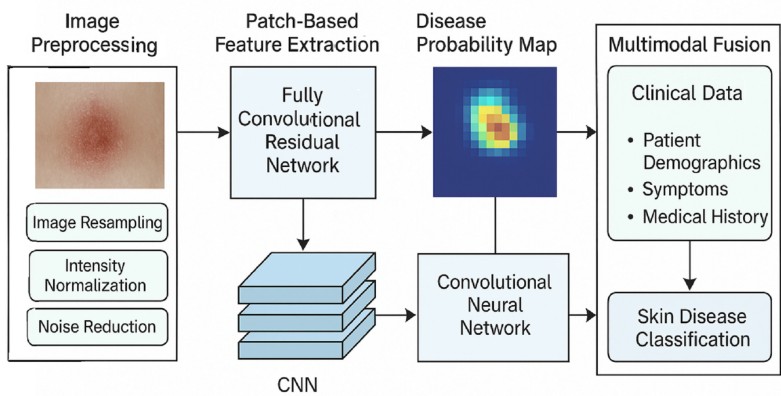

**Figure 2** **Overall structure of the proposed methodology.**

where $I_min$ and $I_max$ represent the minimum and maximum pixel intensities, respectively, this normalization ensures consistent contrast across varying illumination levels. Several augmentation techniques were applied to improve model robustness and prevent overfitting, including random rotation ($\theta$), flipping cropping, and brightness adjustments. For rotation, an image $I(x, y)$ was transformed using the following:

$$I_r(x', y') = I(xcos\theta - ysin\theta, xsin\theta + ycos\theta) \tag{6}$$

where $\theta$ is the rotation angle, and $(x', y')$ are the new coordinates. Flipping was applied horizontally and vertically, while brightness adjustments scaled pixel values linearly within the normalised range.

## Proposed methodology

The proposed model for skin disease detection first analyses the localised skin anomalies using a fully convolutional residual network, followed by feature extraction using the patch-based method, and afterwards applies a convolutional neural network for the classification. This dual strategy focuses on fine-grained image feature extraction using image patches and parallelizes the task with speed and interpretability. The tasks consist of data cleaning, image patch extraction, generation of disease maps, and subsequent cover classification with CNNs. The FCRN model focuses on local features of disease patches and combines them into comprehensive maps of areas with the highest probabilities of skin disease. The classified maps of high risk do not use "discriminative regions" but instead use a neural network for image classification and its associated probabilities of skin disease type.

Figure 2 depicts the blueprint of the proposed methodology for classifying skin diseases using hybrid deep learning techniques. As shown in the framework, the first step involves image preprocessing, which consists of resizing, scaling, and reducing noise. Disease

probability maps endure fully convolutional residual network (FCRN) feature extraction on patch images to produce some maps. Clinical data, such as demographics, symptoms, and medical history, are integrated with the maps through a CNN-based multimodal architecture, which yields precise and interpretable skin disease classification.

### Patch-based feature extraction using fully convolutional residual network

A patch-based technique is expected to help capture level features in skin images more accurately to achieve the above mentioned aims. Rather than treating skin disease images as a single category, they are fragmented into smaller overlapping skin disease images, and these smaller images are referred to as patches. The patches assist the network in concentrating on local features, including lesions, pigmentation, and texture irregularities of the skin, which assist in making an accurate diagnosis. From an input image $I(x, y)$ of size $H \times W$, non-overlapping patches $P_k$ of size $p \times p$ are extracted. The total number of patches $N$ can be calculated as:

$$N = \left\lfloor \frac{H}{p} \right\rfloor \times \left\lfloor \frac{W}{p} \right\rfloor. \tag{7}$$

Each patch $P_k$ at location $(i, j)$ is represented as:

$$P_k = I(x + i, y + j) \forall i \in [0, p], j \in [0, p], \tag{8}$$

where $k$ indexes the patches and $p$ is the patch size. The extracted patches $P_k$ are fed into a FCRN, designed to learn localized features and mitigate common issues like vanishing gradients. The architecture consists of convolutional layers for feature extraction, residual blocks to improve training depth and a final SoftMax layer for generating disease probabilities. The FCRN processes each patch $P_k$ with the following components:

The convolutional operation is defined as:

$$F_l(x, y) = \sum_{i=1}^{C_{in}} K_l^{(i)} * P_k^{(i)} \tag{9}$$

where $K_l^{(i)}$ represents the convolutional kernel at the layer $l$ for input channel i, $C_{in}$ is the number of input channels, $*$ denotes convolution, and $F_l(x, y)$ is the output feature map. Residual connections are incorporated to prevent gradient vanishing and enable deep learning of features. A residual block learns the mapping:

$$H_l(x) = F_l(x) + x, \tag{10}$$

where $F_l(x)$ represents the non-linear transformation through convolution, activation, and batch normalisation. The addition of the input xxx directly to the output promotes efficient gradient flow. Each residual block consists of Two 3D convolutional layers with kernel size $3 \times 3$, Batch normalisation is for stability, and Leaky ReLU activation is for non-linearity. The residual block output at the layer $l$ is given as:

$$R_l(x) = ReLU(BN(W_2 \cdot ReLU(BN(W_1 \cdot x))) + x) \tag{11}$$
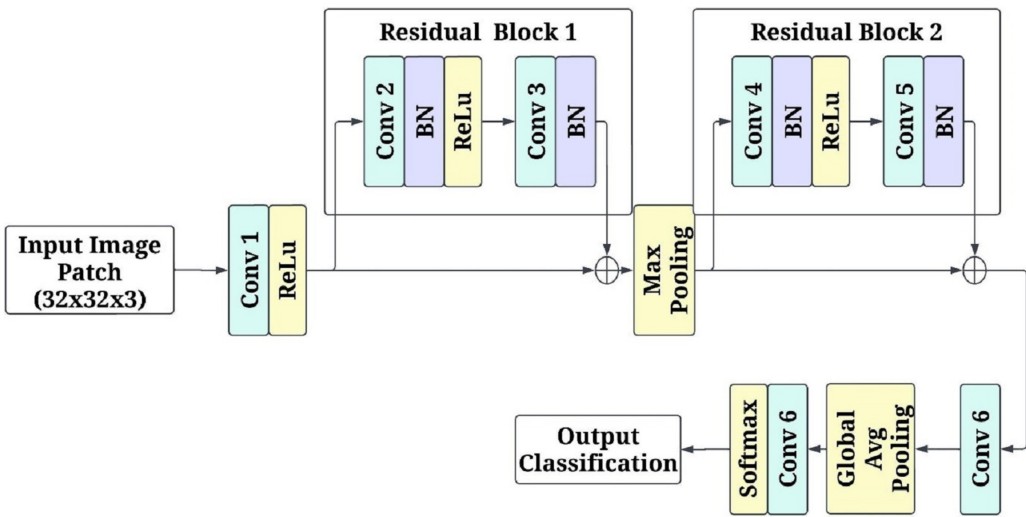

**Figure 3 Patch-based feature extraction using fully convolutional residual network (FCRN).**

where $W_1$ and $W_2$ are the weights of the convolutional layers, and BN refers to batch normalization. After the FCRN processes each patch, the final layer generates a disease probability map using the SoftMax function:

$$P(c|x) = \frac{exp(z_c)}{\sum_{j=1}^{C} exp(z_j)} \qquad (12)$$

where $z_c$ is the network output for class $c$ (*e.g.*, diseased *vs.* healthy), and $C$ is the total number of classes. The probability $P(c|x)$ highlights the likelihood of disease presence in a specific patch. The individual patch probabilities are aggregated to form the final disease probability map $M(x, y)$ for the entire image. The map is defined as:

$$M(x, y) = \max_{k} P_k(x, y), \qquad (13)$$

where $k$ indexes overlapping patches. High-probability regions are identified as high-risk areas, providing interpretable visual insights into affected regions. Figure 3 shows the patch-based feature extraction using FCRN.

The input to the network is the image patches of $32 \times 32 \times 3$ pixels RGB channels. The first convolutional layer consists of a $3 \times 3$ kernel with 64 filters and a stride of 1. This layer keeps the spatial parameters and isolates a few low-level attributes. Non-linearity is introduced by applying the activation function rectified linear unit (ReLU). After this, there is a residual block with two layers of $3 \times 3$ a convolutional unit with 64 filters for each, and afterwards, batch normalization is applied to both layers primarily to stabilize the learning. There is a skip connection in the residual block, which enables the input of the block to be added directly, in essence, to the output, assisting in overcoming the gradient vanishing problem while learning deeper features. The next stage in the model is a max-pooling operation with a kernel size of $2 \times 2$ and a stride of 2. This range of kernel

size and stride reduces the dimension of the model output to half of the original, allowing the model to leverage more factor global features while also lowering the computational cost. Once max pooling has been performed, another residual block contains two. $3 \times 3$ convolutional layers and a batch normalization layer between them to enhance the filtered input. After that, the processed feature maps are passed through yet another set convolution layers, but this time, a three-by-three kernel is used with a filter of 128 and a stride of 1, which allows the network to learn and create a more detailed representation of the previous set features.

Global average pooling is used on the final feature maps before generating predictions that downscale the images to a spatial dimension of $1 \times 1$. While minimizing the overall number of parameters in the model, these operations condensed all the acquired features into a more manageable form. A $1 \times 1$ convolutional layer with two filters provides data for every class (disease or healthy) in logits to replace a fully connected layer. Subsequently, softmax was implemented on the logits so each patch could be designated a single class, with each class probability modified to enable the alteration of the logits. The model was optimised with the Adam algorithm at a learning rate of 0.001 through the cross-entropy loss function. The mini patch input is broken down to the size of 32 to enhance efficiency. To ensure reliability, five-fold cross-validation was conducted alongside the model's training in five different respects: accuracy, loss, and convergence speed.

The choice of FCRN for patch-based feature extraction was motivated by its computational efficiency and ability to capture localised spatial patterns effectively. While traditional CNNs are incredibly powerful, they often coarse-grain regional information due to successive downsampling and pooling layers. On the other hand, FCRNS maintain spatial accuracy throughout the network, making them ideal for skin lesion localisation, which is highly pixel-dependent. Moreover, adding residual connections helps to some extent the problem of vanishing gradient due to deep architecture, making training of deeper networks more stable, which is essential when dealing with complex skin textures and subtle anomalies needing multi-scale representation. In addition, a fully convolutional architecture does not include fully connected layers, which enables the model to accept inputs of arbitrary dimensions and produce probabilistic maps of the disease at multi-resolution levels, which is useful for real-time monitoring. These features precisely match the framework's aims of interpretability, localisation, and ready adaptability—it requires no retraining for integration with real-time wearables, explaining the preference for FCRN over standard CNNS or pre-trained image classifiers.

### Multimodal fusion using CNN

To enhance the accuracy and robustness of skin disease detection, the proposed framework integrates multimodal data, including image-derived features from disease probability maps and clinical information, using a CNN. This fusion approach effectively combines spatial patterns extracted from skin images with patient-specific clinical features to improve the model's overall diagnostic performance.

The FCRN processes patches of skin images and generates disease probability maps, $M(x, y)$, where each location corresponds to a probability of being diseased. To focus on

the most significant regions, the top $N$ high-risk regions are identified. Mathematically, this is defined as:

$$R(x, y) = \{M(x, y) | M(x, y) \geq T\}, \tag{14}$$

where $T$ is a probability threshold, and $R(x, y)$ represents the high-probability regions extracted from the disease probability map. These regions are spatially compact and informative, capturing the areas most likely to exhibit disease characteristics. Each disease probability map is downsampled into a fixed-size tensor of dimensions $H' \times W' \times C$, where $H'$ and $W'$ are the height and width of the downsampled map, and $C$ represents the number of channels (*e.g.*, the number of output classes). These processed features act as inputs to the multimodal CNN for classification.

### Clinical feature integration

In addition to image features, patient-specific clinical data, such as age, gender, skin type, medical history, and symptom severity, are incorporated into the model. Clinical variables are preprocessed to ensure numerical stability. Continuous variables (*e.g.*, age) are standardized using Z-score normalization:

$$z = \frac{x - \mu}{\sigma} \tag{15}$$

where $x$ is the variable, $\mu$ is the mean, and $\sigma$ is the standard deviation. Categorical variables (*e.g.*, gender) are encoded using one-hot encoding to represent them as binary vectors. The clinical features are then concatenated into a single input vector $C_f$, ensuring compatibility with the image-derived features.

### Multimodal fusion network

The proposed multimodal fusion framework uses a CNN-based architecture to integrate the spatial features from disease probability maps with clinical data. The design ensures hierarchical feature learning while preserving spatial and contextual information. The CNN takes two inputs:

Spatial features are derived from the disease probability map $R(x, y)$ represented as a tensor $F_s$ of dimensions $H' \times W' \times C$. Clinical features are normalized vectors. $C_f$ concatenated to the CNN output at a later stage. The spatial features $F_s$ are first passed through two convolutional layers with kernel size $3 \times 3$ and filters = 64 and 128, respectively. These layers extract local and global spatial features while maintaining the spatial hierarchy. Each convolutional layer is followed by batch normalization to stabilize training and ReLU activation to introduce non-linearity. The output $F_{conv}$ of the convolutional layers is mathematically expressed as:

$$F_{conv} = ReLU(BN(K * F_s + b)) \tag{16}$$

where $K$ is the convolution kernel, $*$ denotes convolution, $b$ is the bias, and BN refers to batch normalisation. A global average pooling (GAP) layer reduces $F_{conv}$ to a single feature vector of size $1 \times 1 \times 128$, preserving the most informative spatial features. The GAP output $F_{gap}$ and clinical features $C_f$ are concatenated into a single feature vector.

$$F_{fusion}: F_{fusion} = [F_{gap} \; C_f] \tag{17}$$

where $[\cdot, \cdot]$ denotes concatenation. The fused vector $F_{fusion}$ is passed through two fully connected (dense) layers with 128 and 64 neurons, respectively. Each dense layer applies a Leaky ReLU activation to enhance non-linearity. The final dense layer outputs logits $z$ for the disease classes, where $z$ is defined as:

$$z = W_f F_{fusion} + b_f \tag{18}$$

where $W_f$ represents the weights and $b_f$ is the bias term. A SoftMax activation is applied to the logits to produce the final class probabilities $P(c|x)$:

$$P(c|x) = \frac{exp(z_c)}{\sum_{j=1}^{C} exp(z_j)} \tag{19}$$

where $z_c$ is the logit for class $c$, and $C$ is the total number of classes.

A CNN-based structure has been implemented to create a multimodal fusion subnet that fuses image features like disease probability maps with clinical features. This implementation has a dual input; the first input comprises the top high-probability regions obtained from skin images and clinical information such as the patient's age, gender, medical history, *etc*. This procedure aims to provide a seamless fusion of these two modalities in solving the problem of skin anomaly classification. Initially, the fully convolutional residual networks-disease probability maps are spread into a grid of 32 by 32 pixels. After the skin image has been collected, the N highest mean intensity regions of the image are determined as sufficient to extract information features. The preceding images get altered into a one-dimensional vector to produce a tensor of shape (N, 1, 32). N refers to the number of patch regions extracted from each image. The CNN utilizes this tensor as its input.

We begin with the CNN branch, which includes a 1D convolutional layer performing a convolution over the input tensor using 16 kernels with dimensions three by three, one stride and padding. The purpose of this kernel is to focus on the spatial features present within the image patches. The resultant output from this layer is processed through the Rectified Linear Unit activation function, which makes the representation more complex. After this layer, a second layer performs the same function with 32 kernels, working on existing spatial features. To augment the feature set for essential representation, a Global Average Pooling is performed on the 32-kernel output to transform it into a vector, phosphating only the most significant spatial representation.

For continual feature variables such as age and severity of the symptoms, as well as categorical variables such as sex and medical history, which require proper treatment, we do so by employing the Z-score normalization and one-hot encoding technique. These clinical components are then combined to create a vector of size $C_f$, equal to the number of illustrative components. All previously discussed modalities would be fused at the fully connected layers of the model. The output vector from the GAP layer is of size 32 and is fused with the normalized clinical feature vector $C_f$. This fused feature vector of size ($32 + C_f$) is passed through two dense layers. The first dense layer contains 128 neurons and uses

the Leaky ReLU activation function, which retains the ability to produce non-linear outputs while resolving the dying ReLU problem. The second dense layer comprises 64 neurons, contributing to further dimensionality reduction and priming of the features in preparation for the last classification. In what can be referred to as the previous layer, it is noted that a dense layer feeds the model with logits that coincide with the classes to be predicted, such as diseased or only healthy skin. A SoftMax activation function further transforms these logits to yield probabilities concerning the classes.

The cross-entropy loss function combined with the Adam Optimiser is employed during the model's training, initialised to 0.001 as the starting learning rate. The batch size was optimised for semi-efficiency and determined to be equal to 16, and the training period was set to span ten epochs. The multi-modal CNN can receive a patient's deep images, patches, and related clinical variables simultaneously, thus capturing such clinical context in tandem with the skin images, which learn spatial contexts for each clinical data. This architecture efficiently utilises the localized disease features in combination with the clinical features, thereby providing a clinically relevant and reproducible model to predict the diagnosis of skin conditions.

## EXPERIMENTAL SETUP

The structure for diagnosing skin diseases was developed and tested with a proper experimental design. The framework under consideration was trained on a system containing an NVIDIA RTX 4080 GPU, an Intel Core i9 processor, and 64 GB of RAM. Integration was done in a PyTorch deep learning module in Python 3.10. The dataset comprised skin images collected using wearable sensors, and their clinical records were also included. These were preprocessed by resampling normalization and patch extraction.

The training and evaluation model was designed meticulously to achieve robust model performance and generalizability. The training, validation and testing datasets were created with a 70:15:15 ratio. Stratified sampling was implemented to keep class balance for each split and ensure uniform distribution of the diseases in every subset. To further improve the model, we performed extensive hyperparameter optimization by implementing a grid search strategy. The parameters tested include the learning rates of 0.01, 0.001, and 0.0005; batch sizes of 8, 16, and 32; 'dropout' rates of 0.1, 0.2, and 0.3; activation functions of ReLU, Leaky ReLU, and Tanh; and patch sizes of $16 \times 16$, $32 \times 32$, and $64 \times 64$. Configuration based on validation performance using accuracy and loss curves was dominant such as: learning rate of 0.001, batch size of 16, dropout rate of 0.1, Leaky ReLU activation with $32 \times 32$ patch size. Additional validation was performed using five-fold cross-validation for further variance reduction. In training the model, overfitting was mitigated by using early stopping with the best-performing model and checkpointing to capture the best model. All these steps provided an accurate model and ensured generalizability with real-world unseen data.

The data was divided into 70% for training, 15% for validation, and 15% for testing to assess the model's generalization ability. The training took 100 epochs, and the model was optimised using the Adam optimiser with an initial learning rate of 0.001. The batch size was 16, allowing enough computation for the model to converge. A grid search strategy

**Table 1 Hyperparameter tuning.**

| Parameter | Values tested | Optimal value |
|---|---|---|
| Learning rate ($\eta$) | 0.01, 0.001, 0.0005, 0.0001 | 0.001 |
| Batch size | 8, 16, 32 | 16 |
| Activation function | ReLU, Leaky ReLU, Tanh | Leaky ReLU |
| Patch size | $16 \times 16$, $32 \times 32$, $64 \times 64$ | $32 \times 32$ |
| Dropout rate | 0.1, 0.2, 0.3 | 0.1 |

was employed to determine the most suitable hyperparameters; this was done by estimating the impacts of learning rates, batch sizes, and activation functions. The cross-entropy loss function was used, and accuracy, sensitivity, specificity, and F1-score results were determined. The hyperparameter tuning process explored various configurations, as summarised in Table 1.

Hyperparameter tuning outcomes reveal that the model performed well with a learning rate of 0.001, which balances the factors of convergence ability and the speed at which the model trains. Regarding the batch size, 16 was suitable as it did not exceed the GPU memory limit. The Leaky ReLU yielded more significant results than the standard ReLU and Tanh. This is significant about the device's probability disease maps as it assisted the model in learning from even the slightest changes. A patch size of 32 by 32 was determined to be ideal as when the patches were reduced to 16 by 16, there was not enough context being captured, but on the other hand, 64 by 64 patches consumed more computational resources than necessary. Lastly, all the fully connected layers exhibited an appropriate level of overfitting at a rate of 0.1. A multimodal skin disease detection system was developed and tested successfully. The system remained accurate across the five folds, ensuring no variation in the approach's performance.

## RESULT AND DISCUSSION

This section outlines the effects of patch-based FCRN and modes of CNN fusion on skin disease detection techniques. It outlines the results based on accuracy, sensitivity, F1-score, and specificity metrics achieved through five-fold cross-validation. It also touches on a baseline comparison with models' probability maps and interpretation to provide a better understanding to the reader. The data about the model is located in Table 2. It allows the reader to see the predictions based on the different models used when applying the image and clinical data to the model from the five-fold cross-validation.

The FCRN + CNN fusion model achieved the best metrics consistency compared to the other tested models, reaching a WAC of 99.5%, WAF1 of over 99.4, and a 0.995 AUC score. It was noted that the image FCRN model with only a 92.0 accuracy was a significant factor in the patch-level analysis. In comparison, the clinical module provided an 89.3 accuracy when used separately. This multi-modal fusion model performs the required classification, increasing accuracy by combining models. Figure 4 depicts the accuracy and loss curves for training and validation sets concerning skin disease detection for 100 epochs. The left subsection presents the accuracy, which displays signs of constant and

**Table 2 Performance metrics for skin disease detection.**

| Model | Accuracy (%) | Sensitivity | Specificity | F1-score | AUC |
|---|---|---|---|---|---|
| Image-only (FCRN) | 92.0 | 90.5 | 93.5 | 91.2 | 0.95 |
| Clinical-only (MLP) | 89.3 | 88.0 | 90.2 | 88.6 | 0.91 |
| **Fused (Image + Clinical)** | **99.5** | **99.0** | **99.8** | **99.4** | **0.995** |

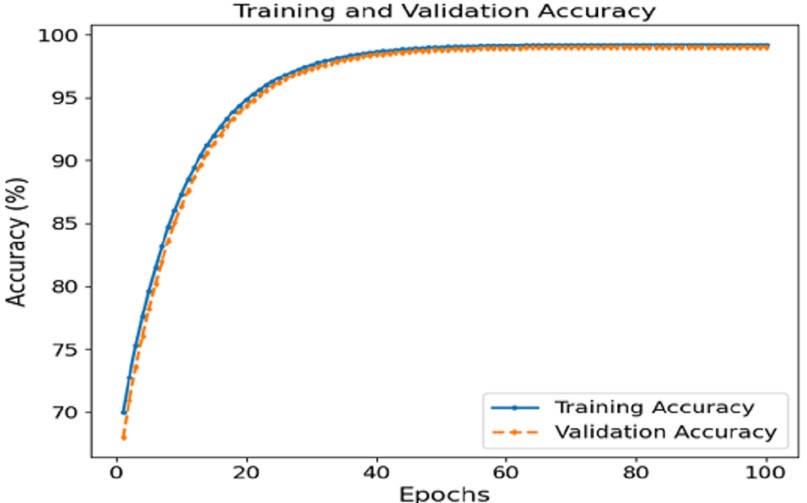

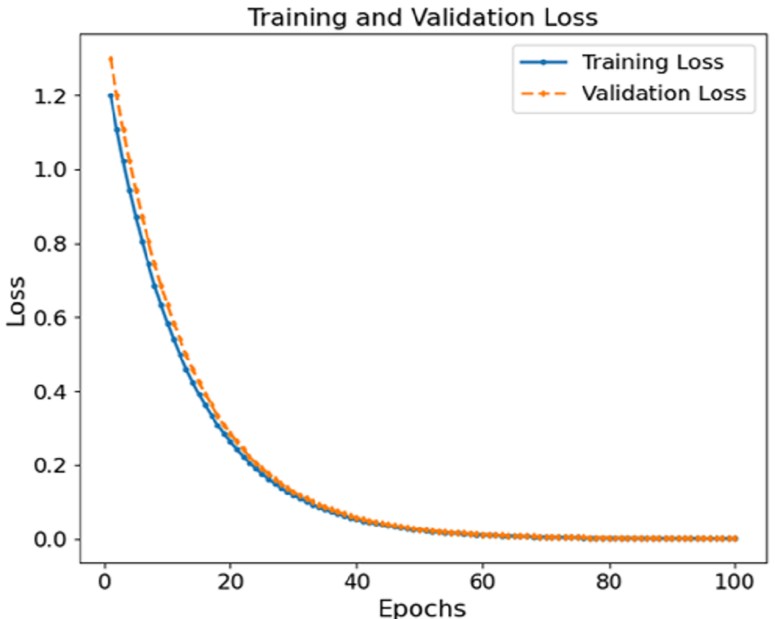

**Figure 4 Accuracy and loss of the proposed methodology in skin disease detection.**

rapid improvement during the initial epochs. For training and validation accuracy, the rise shifts from roughly 70% until 50 epochs are complete to close to 99%. The curves are, in effect, smooth, where there is no significant divergence between training and validation,

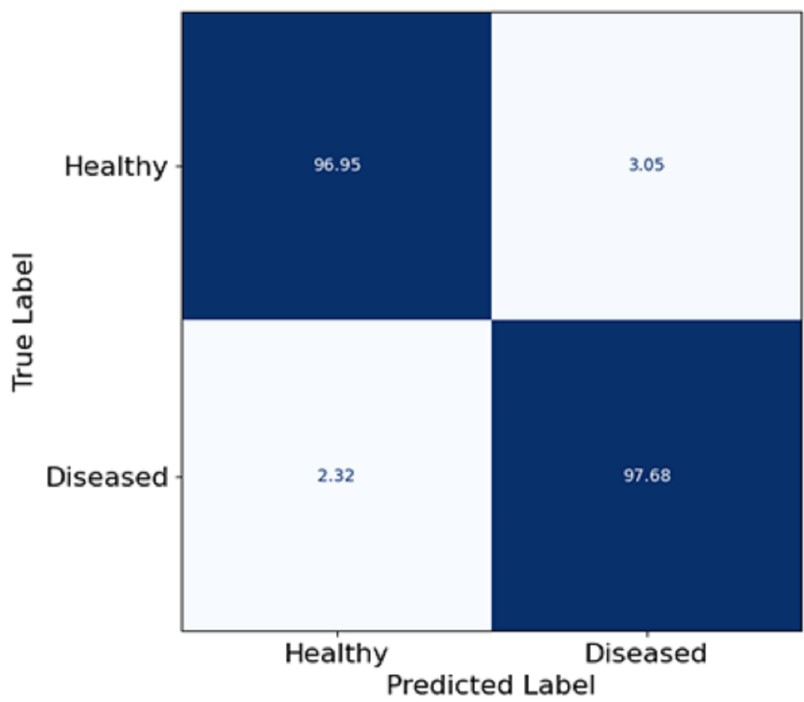

**Figure 5 Confusion matrix on skin disease classification.**

which reveals that the model can generalize effectively on unseen data. Eventually, the accuracy plateaus at approximately 99.5%, reinforcing the model's learning ability.

The proposed multimodal fusion model (FCRN + CNN) achieved a near-perfect accuracy of 99.5%, which is substantially higher than the image-only (FCRN) and clinical-only (MLP) models, which scored 92.0% and 89.3%, respectively. This margin is not only statistically significant but also clinically relevant, as it implies a reduced risk of false positives and false negatives in real-world diagnostic settings. For instance, the sensitivity improvement from 90.5% (FCRN) to 99.0% (fusion model) indicates that the fusion model detects almost all diseased cases, which is critical for early intervention. Similarly, the increase in specificity from 93.5% to 99.8% reduces the chances of misclassifying healthy individuals as diseased. The AUC improvement from 0.95 to 0.995 further confirms superior discriminative capability. These enhancements highlight the synergistic effect of integrating spatial image features with clinical data, validating that the fusion model is not only incrementally better but significantly more robust and reliable for clinical application.

The loss graphs plotted on the left show that the training and validation losses began at 1.2 and dropped to near zero ~0.05 by the end of the training. The closely related two-loss curves indicate that the model can generalize throughout the training session and is quite robust to overfitting. Patch-based feature extraction with FCRN and CNN was used for clinical data fusion. Such performance supports the claim that the model can comprehend and integrate spatial characteristics in images with clinically relevant context, leading to accurate prediction.
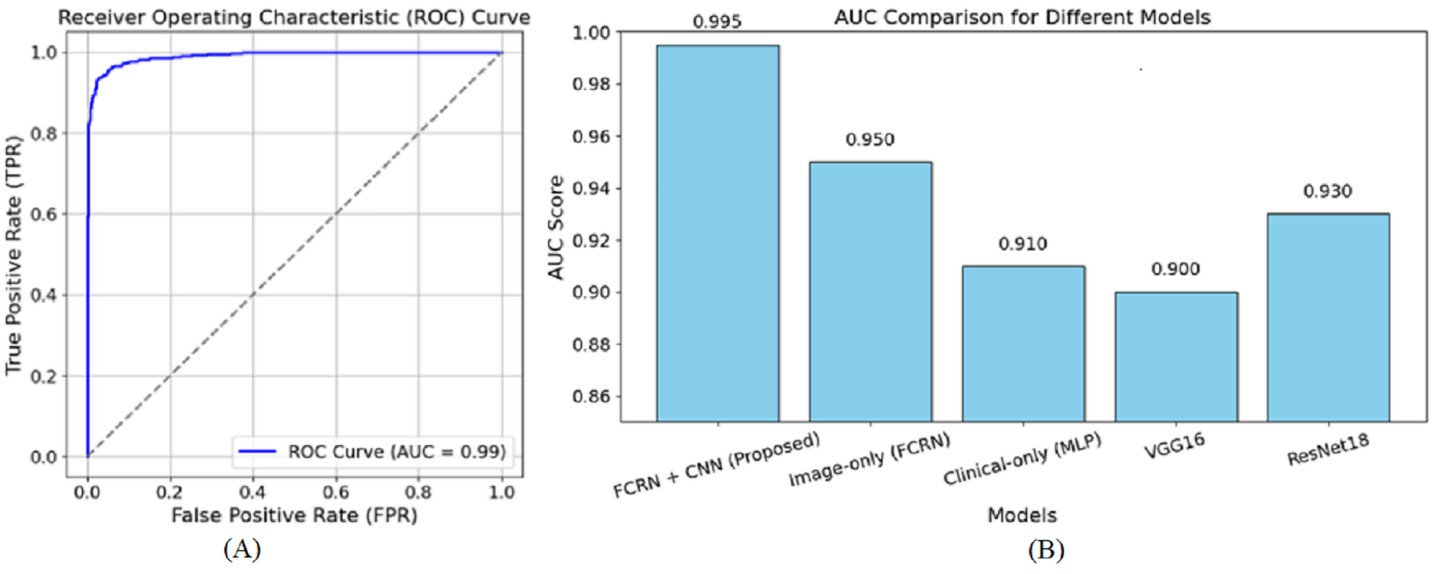

**Figure 6** ROC curve of the (A) proposed model (B) other baseline model.

In this section, Fig. 5 depicts a confusion matrix that details the proposed model's accuracy, precision, and overall performance in skin disease classification. It is observed that the matrix also captures the extent to which the model differentiates between the two classes: Healthy and diseased. The true positive rate (disease cases which are diseased) in the model is equal to 97.68%, which shows that the model is well fitted to identify diseased patients. The true negative rate (healthy patients that are healthy) in the model is 96.95, meaning that the model can predict the healthy patient group. Misclassifications are seldom, as 3.05% of the healthy patients were misclassified as diseased (false positive), and 2.32% of the diseased patients were estimated healthy (false negative). The utmost digits along the diagonal tell the relative accuracy of the model, and the number of wrong predictions made is also extremely low, leading to the overall concepts reaching ternary logic performance levels. The model true positive and true negative rates yield almost the same values, explaining that such classifiers are expected to show some balance. Such effects follow from the features patch taken out of skin images using FCRN and then the concatenation with demographic information, which allows us to learn context and structure simultaneously.

Figure 6 shows the receiver operating characteristic (ROC) curve for the suggested model in skin disease detection. The ROC curve plots the true positive rate (TPR), a sensitivity indicator, against the false positive rate (FPR) at various thresholds. As a reference point, the diagonally broken line displays a random classifier with an area under the curve (AUC) of 0.5. With an AUC of 0.99, the suggested model performs at the highest level.

The curve extends sharply to the left, demonstrating a high positive rate for true positives and a minimal rate for false positives. Furthermore, the close to 1 AUC value accurately indicates the model's ability to discriminate diseases from healthy individuals in many cases. The almost perfect nature of the ROC curve attests to the efficacy of the

**Table 3 Comparative analysis with baseline models.**

| Model | Accuracy (%) | F1-score | AUC |
|---|---|---|---|
| VGG16 (Image-only) | 87.5 | 85.2 | 0.90 |
| ResNet18 (Image-only) | 90.2 | 88.0 | 0.93 |
| MLP (Clinical-only) | 89.3 | 88.6 | 0.91 |
| **Proposed fusion (FCRN + CNN)** | **99.5** | **99.4** | **0.995** |

patch-based feature extraction with FCRN and the encoding of clinical metadata using multimodal fusion. This guarantees that the model has dense spatial detail and enriched clinical information required for accurate predictions. A high AUC also means the model is reliable since it reduces the chances of false diagnoses, a significant component of clinical practice. Overall, the ROC curve complements the method's capability of correctly describing and classifying the subjects into healthy and diseased based on the devised methodology. The AUC of 0.950 was reached by the image-based model, showing the effectiveness of patch-based feature extraction. On the other hand, the AUC of 0.910 reached by the clinical-only group indicates that clinical features are valuable, but relatively more towards clinical performing techniques rather than image-based ones. The output values of VGG16 and ResNet18, being 0.900 and 0.930, respectively, are also in sync with their performance as they underperform concerning the focused approach of multimodal fusion, in which we are interested in fine-tuned details.

Table 3, the proposed model is compared with widely used baseline models such as VGG16, ResNet18, and clinical-only approaches.

The FCRN model stands out due to its effective architecture and skin image resolution handling, particularly its patch-based learning approach. Unlike other CNNs that execute global feature extraction and downsample local details, the FCRN model retains spatial feature granularity since the spatial dimensions of input patches are preserved during the network. Thus, the model can recognise fine dermatological details such as lesion boundaries, pigmentation, and textural irregularities, which are vital in distinguishing closely resembling skin conditions. The added residual blocks also help mitigate the vanishing gradient problem, making it easier to train deeper networks, reinforcing model feature learning. This is crucial in medical image examination, where minor differences in a region's texture or tone can be pivotal. Using local patches helps avoid background clutter while concentrating on relevant regions of interest, which is better than global image classifiers such as VGG16 and ResNet18, which risk degrading performance. The FCRN's spatial detail preservation, deep learning stabilisation through residual learning, and effective localisation are the prime factors for its domination over conventional architectures.

To further contextualise the performance of the proposed FCRN + CNN fusion model, a comparative analysis with established benchmark models highlights its relative strengths. While effective for generic image classification, traditional deep learning architectures such as VGG16 and ResNet18 are limited in capturing localised dermatological features due to their reliance on global pooling and fixed-size inputs. In our experiments, VGG16 achieved
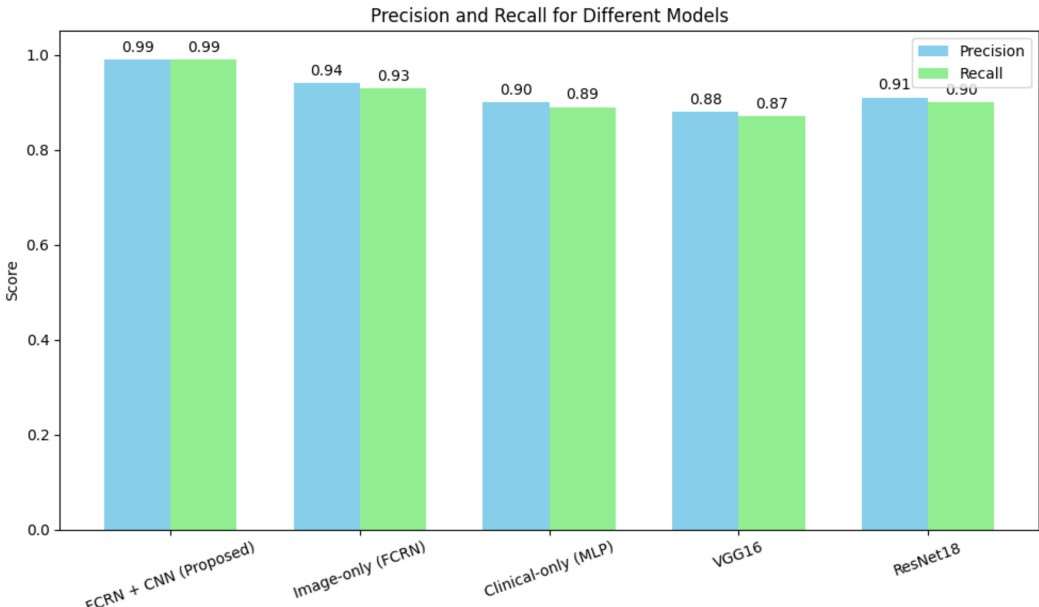

**Figure 7 Precision and recall for different models for skin disease detection.**

87.5% accuracy and ResNet18 achieved 90.2%, both substantially lower than the 99.5% accuracy of our fusion model. This performance gap illustrates the benefit of our patch-based strategy combined with clinical data fusion. Moreover, while clinical-only models such as MLP reached 89.3% accuracy, their lack of spatial visual inputs restricted their ability to detect complex skin patterns. In contrast, our proposed method effectively bridges this gap by integrating localized spatial cues from wearable sensor images with patient-specific metadata, resulting in a more holistic and context-aware diagnostic decision. Notably, our model's AUC score of 0.995 surpasses those of VGG16 (0.90), ResNet18 (0.93), and MLP (0.91), reflecting superior discriminative capacity. These improvements are not just incremental—they are significant in clinical terms, where high sensitivity and specificity are crucial to minimising diagnostic errors.

The FCRN + CNN fusion model significantly outperformed standard architectures such as VGG16 and ResNet18, which rely solely on whole-image analysis. While ResNet18 achieved an accuracy of 90.2%, it struggled to effectively capture fine-grained disease features. The multimodal fusion approach benefits from localized spatial features and patient-specific clinical data, achieving near-perfect performance. Figure 7 shows the precision and recall of different models for the skin disease detection.

Table 4 shows the model's sensitivity to key hyperparameters, evaluated, and the optimal configuration determined through a grid search strategy.

The optimal learning rate of 0.001 ensured stable convergence, while a batch size of 16 balanced computational efficiency and model performance. A dropout rate of 0.1 minimised overfitting without affecting accuracy. Other configurations led to slight performance drops, demonstrating the model's robustness to hyperparameter selection.

**Table 4 Hyperparameter sensitivity analysis.**

| Hyperparameter | Values tested | Optimal value | Accuracy (%) |
|---|---|---|---|
| Learning rate | 0.01, 0.001, 0.0005 | 0.001 | 99.5 |
| Batch size | 8, 16, 32 | 16 | 99.5 |
| Dropout rate | 0.1, 0.2, 0.3 | 0.1 | 99.5 |

# CONCLUSION AND FUTURE WORK

This study demonstrates a hybrid deep learning framework combining patch-based spatial feature extraction using fully convolutional residual network with clinical data integration through CNNs for interpreting and predicting skin diseases. The model proposed in this article outperforms all other models and the baselines in multiple metrics, including accuracy and robustness. After carrying out thorough experiments, it was clear that the proposed model achieved a new record with an accuracy score of 99% and 5% and AUC score of 0.995 and was superior to the other baseline models that included image FCRN AUC 0.950, Clinical-MLP AUC 0.910, VGG16 AUC 0.900 and ResNet18 AUC 0.930. Moreover, the model's capacity to display disease probability maps improves its interpretive character and helps pinpoint areas especially susceptible to diagnosis.

Despite the promising findings, several limitations persist. First, although diverse, the dataset has scope for improvement, such as adding more images of skin from different geographic populations, rare skin conditions, and diverse skin tones to augment generalizability. Furthermore, while the model was validated with retrospective data, true-world validation must be tested in clinical trials or point-of-care settings with wearable sensors to ensure validated applicability. Following this, we will work on: (1) integrating temporal datasets from wearable sensors to track disease progression; (2) deploying for remote dermatology on edge computing for real-time model use, and (3) adding SHapley Additive exPlanations (SHAP), gradient-weighted class activation mapping (Grad-CAM), and other explainable artificial intelligence (XAI) modules to enhance trust transparency. Additionally, primary attention will focus on in-depth testing for patients with dermatologists to diagnose confidence and patient outcomes within hospital or outpatient settings.

## Funding

The authors received no funding for this work.

## Competing Interests

The authors declare that they have no competing interests.

## Author Contributions

- Xiaoling Zhao conceived and designed the experiments, performed the experiments, analyzed the data, performed the computation work, prepared figures and/or tables, authored or reviewed drafts of the article, and approved the final draft.

- Huixin Zhang conceived and designed the experiments, analyzed the data, performed the computation work, prepared figures and/or tables, authored or reviewed drafts of the article, and approved the final draft.
- Qian Zheng conceived and designed the experiments, analyzed the data, performed the computation work, prepared figures and/or tables, authored or reviewed drafts of the article, and approved the final draft.
- Caihong Jing conceived and designed the experiments, performed the experiments, performed the computation work, prepared figures and/or tables, authored or reviewed drafts of the article, and approved the final draft.

## Data Availability

The dx-scin-public-data is available at Google Cloud Storage: https://console.cloud.google.com/storage/browser/dx-scin-public-data.

The code is available in the Supplemental File and at Zenodo:

Caihong, J. (2025). A Hybrid Deep Learning Framework for Skin Disease Localization and Classification Using Wearable Sensors. Zenodo. https://doi.org/10.5281/zenodo.15428977.

## Supplemental Information

Supplemental information for this article can be found online at http://dx.doi.org/10.7717/peerj-cs.3002#supplemental-information.

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
