# Peer review of "A hybrid deep learning framework for skin disease localization and classification using wearable sensors"

_PeerJ Computer Science, doi:10.7717/peerj-cs.3002_

## Round 0.1 · original submission · Major Revisions

Please address the reviewer comments.

**Language Note:** The review process has identified that the English language must be improved. PeerJ can provide language editing services - please contact us at [email protected] for pricing (be sure to provide your manuscript number and title). Alternatively, you should make your own arrangements to improve the language quality and provide details in your response letter. – PeerJ Staff

Reviewer 1 ·

Basic reporting

* The manuscript: Clear and unambiguous, professional English language used throughout; The manuscript generally uses clear and professional language; however, some sections (e.g., the explanation of the convolutional neural network-based fusion model) could benefit from improved clarity and conciseness (Page 8).

** Intro & background to show context; The introduction provides sufficient background on skin diseases and their classification. However, additional references to recent studies on deep learning-based skin disease classification would strengthen the literature review (Page 4).

*** Structure conforms to PeerJ standards; The manuscript follows a logical structure, but the presentation of Figures 2 and 3 could be improved for better clarity (Page 11).

**** Does the Introduction adequately introduce the subject and make it clear what the motivation is?.... The motivation is clear, but it would help to emphasize the limitations of previous approaches more explicitly (Page 5).

Experimental design

*Article content is within the Aims and Scope of the journal; The content aligns with the journal's aims, focusing on AI-based medical image analysis.

** Rigorous investigation performed to a high technical & ethical standard; The methodology is sound, but more details are needed on the training/validation split and hyperparameter tuning (Page 14).

*** Methods described with sufficient detail and information to replicate; The implementation details are well-explained, but the rationale behind certain design choices, such as the selection of the fully convolutional residual network (FCRN), needs clarification (Page 10).

**** Is there a discussion on data preprocessing and is it sufficient/required? ... Data preprocessing steps, including intensity normalization and noise reduction, are described well (Page 9).

***** Are the evaluation methods, assessment metrics, and model selection methods adequately described?... The evaluation methods (accuracy, sensitivity, specificity) are explained, but further discussion on the significance of differences between models is needed (Page 15).

******Are sources adequately cited? Quoted or paraphrased as appropriate? ... Yes, sources are adequately cited.

Validity of the findings

* Are the experiments and evaluations performed satisfactorily? ... The experiments are well-designed, but additional insight into why FCRN outperformed other models would strengthen the findings (Page 18).

** Is there a well-developed and supported argument that meets the goals set out in the Introduction?... The argument is well-developed, but additional comparative analysis between the proposed model and existing benchmarks would be helpful (Page 19).

*** Does the Conclusion identify unresolved questions/limitations/future directions?... The conclusion addresses limitations but could suggest more specific future directions, such as expanding the dataset or testing in clinical settings (Page 21).

Additional comments

* Improve the clarity of Figures 2 and 3 to enhance the interpretation of the data (Page 11).

** Expand the literature review to include more recent and relevant studies on deep learning in medical imaging (Page 4); for example:
- Ahmed, M. A., et al. "Automatic COVID-19 pneumonia diagnosis from x-ray lung image: A Deep Feature and Machine Learning Solution." Journal of Physics: Conference Series. Vol. 1963. No. 1. IOP Publishing, 2021..

*** Clarify the motivation behind using the FCRN and CNN-based fusion model (Page 10).

**** Provide more details on the training/validation split and model performance variability (Page 14).

·

Basic reporting

The paper is written in clear and professional English. The grammar, vocabulary, and overall language style are suitable for an academic paper. The Introduction gives an overview of the significance of skin disease detection, the limitations of traditional methods, and how deep learning/wearable sensors can address these challenges. The paper follows a typical research article structure: Introduction, Related Works, Methods, Experimental Setup, Results & Discussion, and Conclusion. The Introduction explains general aspects of skin diseases, the need for early and accurate detection, and how wearable sensors combined with deep learning offer a promising solution. The motivation for combining image-based features with clinical data is well articulated. Also, the contribution of the work is acceptable.

In addition to all of that, another overall language check is required (still has some typos like line 184, balance the two). Also, check all the subscripts and superscripts in equations 1 through 19.

Experimental design

The paper focuses on a computational approach to skin disease detection using deep learning and wearable sensor data. The methodology is thorough, the authors described data preprocessing, patch-based feature extraction, model architecture, training procedures, and evaluation metrics. Also, the authors described data preprocessing steps (resampling, normalization, noise reduction), model architecture (Fully Convolutional Residual Network + CNN fusion), and hyperparameters (learning rate, batch size, dropout rate). They also provide the link/DOI to the dataset as well as the code. The paper included subsections on preprocessing, covering image resampling, intensity normalization, and segmentation. These steps are well justified given the variability in images from wearable sensors.

Please add a detailed block diagram for the overall multi-modal framework in section 3.3. or 3.3.3.

Validity of the findings

The authors report accuracy, sensitivity, specificity, F1-score, and AUC. They compare the proposed method to baseline models (VGG16, ResNet18, etc.) and use five-fold cross-validation. Hyperparameter tuning is also described, indicating a robust evaluation metric. The paper included references to previous work on machine learning, skin disease detection, and wearable sensors. Citations appear appropriate, with references supporting statements in both the Introduction and Related Works sections.

---

## Round 0.2 · accepted · Accept

Dear authors, reviewers report that you have solved all the previous concerns. As reviewers, the revised manuscript has been thoroughly reviewed, and all prior concerns have been addressed with professionalism and clarity. The paper now demonstrates clear and precise language throughout, a well-structured and contextually grounded introduction, and a significantly strengthened and up-to-date literature review.

Reviewer 1 ·

Basic reporting

The authors have addressed the prior concerns with professionalism and completeness. The manuscript now exhibits:
- Clear and unambiguous language throughout.
- A comprehensive introduction that outlines the context and significance of the problem, as well as the motivation for the proposed work.
- A strengthened literature review with updated references to recent work on deep learning for dermatological diagnosis (e.g., Oztel 2023, Muhaba 2022, Sadik 2023), successfully situating the paper within current developments.
- The structure aligns well with PeerJ standards, and figures have been enhanced for better clarity (notably Figures 2 and 3).
- Limitations in prior studies have been well discussed, justifying the need for the proposed framework that integrates clinical metadata and localized feature extraction.

No further comment.

Experimental design

No comment – All aspects related to experimental setup, hyperparameter tuning, dataset splits, and methodological justifications have been clearly detailed.

Validity of the findings

The authors convincingly validate their findings through:
- A thorough comparative analysis with standard models (VGG16, ResNet18, and MLP), showing consistent and significant improvements.
- Quantitative metrics (accuracy, sensitivity, specificity, and AUC) that demonstrate both statistical and clinical relevance.
- Clear reasoning for the superior performance of their FCRN+CNN fusion model.
- An expanded discussion of limitations and future directions, including plans for real-world deployment, SHAP/Grad-CAM integration, and broader population validation.

No further comment.

Additional comments

This is a well-crafted and well-revised manuscript. The authors have responded thoroughly to prior reviewer concerns and improved the clarity, depth, and rigor of the work. The framework is timely, technically sound, and well-positioned for real-world clinical application. Minor editorial polishing may be beneficial, but does not impact scientific content.

·

Basic reporting

After reading the paper thoroughly, it seems to me that the authors have made all the required corrections and suggestions; therefore, I have nothing more to say.

Experimental design

After reading the paper thoroughly, it seems to me that the authors have made all the required corrections and suggestions; therefore, I have nothing more to say.

Validity of the findings

After reading the paper thoroughly, it seems to me that the authors have made all the required corrections and suggestions; therefore, I have nothing more to say.